# Brogidirsen and Exon 44 Skipping for Duchenne Muscular Dystrophy: Advances and Challenges in RNA-Based Therapy

**DOI:** 10.3390/genes16070777

**Published:** 2025-06-30

**Authors:** Annie Tang, Toshifumi Yokota

**Affiliations:** Department of Medical Genetics, University of Alberta, 116 St. & 85 Ave., Edmonton, AB T6G 2R3, Canada

**Keywords:** duchenne muscular dystrophy, exon skipping, exon 44, antisense oligonucleotide, brogidirsen, clinical trial, dystrophin, NS-089/NCNP-02, splice modulation, antisense therapeutics

## Abstract

Duchenne muscular dystrophy (DMD) is a severe inherited muscle-wasting disorder that is associated with severe morbidity and mortality globally. Current treatment options have improved the quality of life of patients, but these treatments are only palliative. There is a need for more DMD treatment options. Antisense oligonucleotide (ASO) therapies have emerged as a promising personalized treatment option for patient groups that possess specific mutations. A subset of these therapies can skip over frame-disrupting exons in the *DMD* gene and can partially restore dystrophin production for individuals with DMD. One novel exon skipping therapy currently being investigated is brogidirsen, an exon 44 that targets ASO using a novel dual-targeting approach. This article will provide an overview of brogidirsen’s history and current clinical trial developments. It will summarize how this investigational therapy compares with other pre-clinical and clinical trial-stage ASO therapies targeting exon 44. Current advances and challenges faced by RNA-based therapies will also be discussed. Overall, brogidirsen is a promising potential addition to existing DMD treatment options, with its clinical trial results showing expression levels above that of the maximum amount of dystrophin expression achieved by current FDA- and EMA-approved exon-skipping DMD therapies. Further research will be needed to determine its overall efficacy and ability to overcome the known limitations faced by other existing ASO therapies.

## 1. Introduction to Duchenne Muscular Dystrophy and Treatments

### 1.1. Duchenne Muscular Dystrophy (DMD) Pathology

DMD is a progressive X-linked neuromuscular disorder that affects 19.8 out of every 100,000 live male births worldwide [1,2]. It is caused by mutations in the *DMD* gene that lead to a lack of functional dystrophin, an important cytoskeletal protein [3]. Unlike the *DMD* gene mutations that result in the milder Becker muscular dystrophy (BMD), DMD is caused by loss of function *DMD* gene mutations that shift the translational reading frame [4]. This results in a loss of functional dystrophin, which leads to functional muscle tissue being gradually replaced with fibrofatty tissue [5]. The large rod-shaped dystrophin protein plays an essential role in maintaining the muscle–cell membrane by linking the subsarcolemmal actin cytoskeleton to the inner part of the sarcolemma [3]. A lack of dystrophin disrupts the subcellular signaling of myofibers, which results in increased muscle membrane stress and damage, ultimately leading to progressive skeletal and cardiac muscle degeneration [6]. This results in a loss of ambulation and eventual death from respiratory or cardiac complications in the thirties to forties, even with current life-extending treatments [7,8]. The standard steroid treatments for DMD aim to slow disease progression, but there is an unmet need for more effective treatments [9].

Current palliative therapies have extended the lifespan of some DMD patients from having an untreated natural history of around 20 years to 30–40 years with treatment [10]. Spinal surgeries and nocturnal ventilation support technologies have significantly improved survival rates [11]. While effective in slowing disease progression, the standard corticosteroid treatments also have severe side effects, such as osteoporosis, cataracts, weight gain, and behavioral changes [11,12]. While these side effects are monitored and included in care considerations, further advancements in the field are required. Some current strategies aim to restore dystrophin levels through gene delivery, CRISPR-Cas9 delivery systems, or exon skipping. Only 20% of regular dystrophin expression levels are needed to reduce the most severe DMD symptoms and produce a milder disease course [13]. Yet, this level of dystrophin restoration in patients has not yet been achieved with any of the above mentioned treatment strategies, indicating a need for continued efforts to improve DMD therapies.

Symptoms of DMD are often seen around the ages of 3 to 4 years through clinical manifestations such as the Gower’s maneuver, an abnormal waddling gait, frequent falls, clumsiness, and toe walking [14]. However, many cases can be diagnosed earlier through recognition, examination and investigation of any male child who is not walking independently by age 18 to 24 months. A diagnosis should be confirmed as early as possible to allow timely genetic counselling for the parents and wider family, A diagnosis is typically confirmed by around the age of 5 years [15]. Diagnostic tests include electrocardiograms, creatine kinase tests, gene testing, and muscle biopsies [16]. Elevated creatine kinase levels are typically tested early, and genetic tests are then used to confirm the diagnosis by identifying the location, type, and pathogenicity scale numbering of mutations within the 79 exon *DMD* gene [17]. Around 70% of DMD patients have deletion(s) or duplication(s) spanning one or multiple exons in the *DMD* gene, which can be readily detected using genetic tests [14]. Muscle biopsies are then used to confirm the absence of dystrophin [14]. Interestingly, female carriers of DMD can exhibit elevated creatine kinase as well [18]. This usually presents as a shifted distribution of creatine kinase levels compared with the normal range, which can be markedly raised in some clinically symptomatic manifesting carriers. This disease affects ambulation, causing difficulties with jumping, running, climbing stairs, and standing up from the ground [2,14]. Treatment care plans shift between the ambulatory and non-ambulatory stages of the disease, but rehabilitation management and glucocorticoid use remain staples throughout the lifespan [14].

The severity of DMD worsens with age, even with current pharmacological, surgical, and holistic strategies [19]. Scoliosis, cardiomyopathy, and respiratory issues usually begin to develop in the teens [19]. The onset of scoliosis and a loss of ambulation typically occurs around 10–14 years [20,21]. Some DMD patients also experience co-morbid cognitive impairment or behavioral conditions, including autism spectrum disorder, obsessive-compulsive disorders, anxiety, intellectual disability, or attention deficit hyperactivity disorder [22]. Around half of DMD patients have a learning disability [23]. This disease has further been associated with sleep disturbances, gastrointestinal problems, chronic pain, stiffness, chewing problems, obesity, cardiac disease, and mental health issues [24]. The epidemiological burden and frequency of DMD are not linked to any demographic, with different populations generally having similar levels of risk. However, differences in age at diagnosis and treatment plans vary globally and between different demographics [25]. The economic, social, and healthcare utilization costs associated with DMD rise as the disease progresses, with the morbidity and mortality of the condition placing a significant burden on healthcare systems worldwide [26].

### 1.2. Antisense DMD Treatments

Antisense oligonucleotides (ASOs) are synthetic nucleic acid-based analog drugs that have shown therapeutic potential. Current ASOs approved by the United States Food and Drug Administration (FDA) used to treat DMD and other diseases include viltolarsen, casimersen, golodirsen, eteplirsen, nusinersen, tofersen, and many others. When used to treat DMD, ASO therapies aim to restore the production of partially functional dystrophin to reduce muscle degeneration and slow disease progression [27]. Sarepta has targeted different exons of the *DMD* gene by developing eteplirsen (exon 51), golodirsen (exon 53), and casimersen (exon 45) [28,29,30]. Nippon-Shinyaku and NS Pharma have progressed viltolarsen (exon 53) into an FDA-approved treatment and are currently conducting clinical trials on brogidirsen (exon 44) [31,32].

A subset of ASO therapies aims to use exon skipping to slow the progression of DMD. As there are in-frame and out-of-frame pathogenic DNA mutations, exon-skipping may require more than one exon to be skipped to return to in-frame reading of the transcribed RNA [33]. Exon-skipping ASO therapies intend to skip over specific segments, known as exons, of the *DMD* gene that are mutated. This restores the open reading frame of the *DMD* gene, which can then lead to the production of a truncated version of dystrophin [34]. These shortened dystrophin proteins are partially functional and are thought to be able to contribute to the dystrophin–glycoprotein complex to provide structural support and stability to the sarcolemma. These exon skipping therapies target pre-mRNA, a stage where the mRNA is not yet ready for translation into dystrophin protein. They do not necessarily target the mutated region of the *DMD* gene, as they can instead target flanking parts that induce the skipping of those mutated parts [35]. The overall aim is to convert a DMD phenotype into a milder BMD phenotype by using ASOs to alter pre-mRNA splicing in a manner that results in the production of a partially functional dystrophin protein [34].

Exons within the DMD gene can be one of 3n, 3n + 1, or 3n − 1 nucleotides; thus, exon deletions, as the most common type of dystrophin mutation [33], can have three different effects on the reading frame. Adding in a ‘deletion’ using exon skipping can only correct the reading frame if the ‘deletion’ complements an existing out-of-frame deletion. For example, a 3n + 1 type of deletion would only be able to restore the reading frame for certain mutations. Inducing the deletion of exon 44, in this case, can then only compensate for deletions of the 3n − 1 type. This is represented by different exon shapes in diagrams, with some potential exon skipping deletions resulting in two exons that do not fit together into a functional reading frame (Figure 1). These are factors to consider when designing ASO exon skipping therapies, along with the exon skipping efficiency, protein functionality, and potential applicability to other mutations [33].

The *DMD* gene has many exons that can be targeted by ASOs. In DMD patients, deletions are frequently found in a 3′ mutational hotspot between exons 43 and 55, with some studies narrowing the region down to exons 45 to 55 [36,37]. Deletions of exons 43, 45, and 52 are common [38]. Another mutational hotspot has been identified within exons 3 to 9 and is known as the 5′ mutational hotspot [39]. Exon-skipping ASOs are designed to restore the mRNA reading frame by skipping the mutated regions. These ASOs can be used to treat patients with deletions, duplications, or small mutations in the *DMD* gene [40]. Single-exon skipping is most applicable to patients with deletions in the rod domain of dystrophin [40]. This protein domain region is more tolerant to internal deletions because the region contains several repeated spectrin-like domains and hinges [2]. By targeting specific mutations within DMD, ASO therapies offer a personalized treatment option for patients with eligible, amenable mutations.

ASOs can be structurally classified into Locked Nucleic Acid (LNA), 2′-O-methyl RNA backbone (MOE), and phosphorodiamidate morpholino oligomer (PMO) subclass groupings [41]. LNA molecules are locked into a single conformation because they contain a bridge between sugar moieties and the phosphate backbone or the furanose ring [42]. MOE-modified molecules are more resistant to nucleases than unmodified RNA, mainly because they substitute a 2′-ethylene glycol in for the ribose 2′OH normally found in RNA [43]. Currently, PMO therapies have dominated the FDA-approved DMD landscape and have been further developed than LNA or MOE-based therapies for DMD. All four conditionally approved DMD-treating ASOs (eteplirsen, viltolarsen, casimersen, and golodirsen) are PMOs [44]. These PMO therapies differ from regular RNA/DNA through sugar and phosphate modifications. They have a backbone made of morpholino ring structures instead of the regular ribose backbone. These morpholino rings are then connected via phosphorodiamidate linkages instead of the regular charged phosphodiester inter-nucleoside linkages [45]. The synthesis of PMOs starts with opening the five-membered ribose rings within ribosides using oxidation [46]. These rings are then closed using ammonia to generate a substituted six-member morpholine ring [46]. Phosphorodiamidate linkages are then added to replace phosphodiester bonds [46]. These phosphorodiamidate linkages result in high metabolic stability, enzymatic resistance, and aqueous medium solubility [47]. PMOs also lack a carbonyl group usually present in nucleic acids, which confers resistance against attacks by proteases, nucleases, and esterases in biological serums [48]. These molecules are also highly resistant to hydrolases [48]. These properties result in few interactions with biological molecules in general, but an ability to bind strongly to target sequences [49].

PMOs are considered a promising treatment avenue for several diseases because of their ability to increase or decrease gene expression through a steric hindrance or splice modulation mechanism [49,50]. They have neutral charges and are used extensively for antisense gene knockdown applications in cancer and developmental biology [45]. ASO treatments for DMD can use the splice modulation mechanism to rescue gene expression [51]. PMOs can bind to complementary sequences through Watson–Crick base pairing with target pre-mRNA sequences at or near exon–intron junctions to alter mRNA splicing [52]. This can exclude exons with mutations that disrupt the reading frame and result in the production of an internally truncated version of dystrophin from the patient’s own genes. Other mechanisms, such as gene therapy and CRISPR-Cas-9 delivery mechanisms, exist to restore function, as well, but have either not progressed past the pre-clinical stage or are currently undergoing evaluations of their efficacy and immunogenic effects [53]. Immunogenic reactions for ASO therapies were a major concern for first-generation ASOs, but structural modifications have decreased the risks for immunogenicity for current ASO treatments [54]. These improvements have enhanced the therapeutic potential and safety profile of ASO therapies, supporting their continued development as a viable approach for treating genetic disorders such as DMD.

This paper will summarize information on and present a comparison between brogidirsen (NS-089/NCNP-02) and other exon 44 skipping therapies in development. It will center on the therapeutic potential of exon 44 skipping and trace back recent developments for exon skipping candidates in the DMD oligonucleotide therapeutic field. Despite the narrow scope of this paper, the comparison presented is useful because it illustrates how certain design features can influence the clinical attributes of the oligonucleotide drug. The design of a single linear PMO sequence targeting two discrete internal splice regulators with exon 44 appears to add sufficient potency to offset any potential need for conjugation to a targeting antibody or cell-penetrating peptide, as is performed with the other exon 44 skipping oligos in clinical development.

## 2. Exon 44-Skipping Therapies

### 2.1. Mechanism and Targets

Currently, there are no federally approved exon 44 skipping therapies. Oligonucleotide-mediated skipping therapies that target exon 44 have the potential to treat up to ~7% of DMD patients [55]. This is due to their ability to address mutations that frequently cluster in regions around exon 44 of the *DMD* gene. Stretching from exon 43 to 55 is a mutational hotspot within the *DMD* gene, with the most common deletion in the *DMD* gene being exon 45 [34,37]. Given the relatively low number of DMD patients with exon 44–45 deletions and a link between this deletion and a milder phenotype, exon 44 skipping in DMD boys who already have an exon 45 deletion or another exon 44-skipping amenable deletion is thought to result in the restoration of the reading frame and yield a shortened but partially functional dystrophin protein [34]. Clinicians identify patient candidates who are amenable to exon 44 skipping through genotyping. Exon 44 skipping may be therapeutically applicable to patients with deletions encompassing exons 45–54, only exon 45, a subset of deletions ending at exon 43, and many other forms of deletions [34]. While not all deletions ending at exon 43 are amenable, applicable ones include some deletions of DMD exons 14–43, 19–43, 30–43, 35–43, 36–43, 40–43, and 42–43 [34,40]. The goal of these therapies is to enable the production of a truncated dystrophin protein that retains partial function, like that present in some BMD patients.

Several ASOs that target DMD exon 44 are currently in the clinical trial stage. Avidity Biosciences is developing AOC1044, Entrada Therapeutics has advanced ENTR-601-44, and NS Pharma is conducting clinical trials for brogidirsen (NS-089/NCNP-02). Currently, the NS Pharma candidate has advanced the furthest in the pipeline, having undergone Phase I/II or II trials in DMD patients. Sarepta Therapeutics, Wave Life Sciences, PepGen, and Dyne Therapeutics are also developing exon 44-skipping therapies that are currently in the pre-clinical stage. These efforts will be detailed below and reflect a growing pipeline of next-generation exon-skipping therapies that integrate novel oligonucleotide designs and delivery platforms aimed at improving tissue uptake and exon skipping efficiency.

### 2.2. Current Pre-Clinical and Clinical Trial Results for AOC1044 and ENTR-601-44

AOC1044, also known as delpacibart zotadirsen, by Avidity Biosciences is an exon 44-skipping therapy that consists of four PMO copies attached to a humanized monoclonal anti-transferrin receptor 1 (TfR1) antibody via linkers [56,57]. The antibody–PMO conjugation is meant to better target the PMOs to muscle tissue, as the TfR1 target is ubiquitously expressed on skeletal, cardiac, and smooth muscle cell membranes [58]. In preclinical humanized mdx mouse models of DMD with an exon 45 deletion, this therapy has been observed to restore up to around 20% of dystrophin in skeletal muscle and 6% in cardiac muscle [59]. The dystrophin restoration levels in cardiac muscle are particularly interesting. This is because existing therapies are inefficient at restoring dystrophin in cardiac tissue, which is an area that needs to be targeted to reduce the effects of cardiomyopathy [59,60]. The safety and tolerability of this therapy were tested in a placebo-controlled, randomized, double-blind EXPLORE44 Phase I/II clinical trial on 40 healthy human volunteers (NCT05670730). The results showed that the AOC1044-treated group only had mild or moderate adverse events, indicating that the therapy was well tolerated [56]. Dose-dependent increases in exon-skipping efficiency were also observed after administering a single 5 mg/kg or 10 mg/kg dose [56]. Currently, this therapy is being tested on DMD patients in the second part of the EXPLORE44 trial.

ENTR-601-44 consists of an exon 44-skipping PMO conjugated to a proprietary peptide that serves as an Endosomal Escape Vehicle (EEV). Entrada Therapeutics has multiple EEV-conjugated therapies under development, and it has optimized the EEV technology to enhance intracellular delivery efficiency into specific organs and tissues [44,61]. The EEV contains cell-penetrating peptides that induce the budding and collapse of endosomes [62]. This helps ENTR-601-44 escape endosomal entrapment and increases its effectiveness [63]. Preclinical studies in D2-mdx mice showed exon skipping in both skeletal and cardiac tissue [64]. In the United Kingdom, the safety and tolerability of ENTR-601-44 were examined in healthy male volunteers aged 18 to 55 from August 2023 to October 2024 in a placebo-controlled study (ISRCTN36174912). Four dose levels were tested, with regular blood and urine tests to monitor kidney function and platelet levels [65]. This therapy had a two-year FDA clinical hold lifted in February 2025 and will begin testing in the United States in the ELEVATE-44-102 trial, a randomized double-blind placebo-controlled Phase 1b multiple ascending dose clinical study [61]. This trial is set to begin in 2026 and will administer ENTR-601-44 at doses ranging from 0.16 mg/kg to 1.28 mg/kg to around 32 ambulatory and non-ambulatory DMD patients [61].

## 3. Brogidirsen Studies and Trials

### 3.1. Preclinical Study Results

Brogidirsen (NS-089/NCNP-02) is an unmodified PMO with a novel dual-targeting approach for DMD exon 44 [32,66]. This therapy obtained orphan drug destination and breakthrough therapy status in the United States and Europe during 2023 [67]. The 24-nucleic acid sequence of NS-089/NCNP-02 is ‘TTGTGTCTTTCTTCTGTTAGCCAC’ (FDA UNII Code: 4KM8P975UM). Brogidirsen contains two directly connected 12-mer sequences that are designed to skip exon 44 by targeting positions 64–75 and 92–103 within the exon (Figure 2) [68]. It has shown therapeutic potential in a previous preclinical study, where a 1 h incubation treatment of patient-derived MYOD-converted fibroblasts possessing a DMD exon 45 deletion resulted in an exon 44-skipping EC_50_ of 0.63 μmol/L 7 days after incubation with the NS-089/NCNP-02 treatment. In a following clinical trial by Nippon-Shinyaku and NS Pharma (NCT04129294), treatment was shown to restore dystrophin expression to approximately 25% of the levels observed in healthy controls [66,68]. Once NS-089/NCNP-02 binds to the targeted sites on RNA, frame-shifting exons can be spliced out and skipped over (Figure 3). Since PMOs lack charge, they are perceived to have a better safety profile and improved stability compared to charged ASOs [44,69].

Preclinically, NS-089/NCNP-02 was identified out of several single ASOs that contained two exon-targeting sequences [70]. Out of 26 22-mer PMOs, high exon 44 skipping levels were observed for molecules targeting the 11–52 and 61–122 positions [70]. Through further screening, oligomers that targeted positions 21–33, 61–73, and 91–103 were shown to have high skipping activity [70]. 24-mer and 26-mer candidates were identified to have the highest levels of activity, and the 24-mer molecule was selected for further study [70]. This 24-mer molecule was NS-089/NCNP-02, and testing NS-089/NCNP-02 in human rhabdomyosarcoma cells showed a dose-dependent increase in exon skipping efficiency when measured through RT-PCR [70].

This treatment was then tested in patient-derived DMD myotubes with a deletion in exon 45 that was amenable to exon 44 skipping [70]. Increased levels of dystrophin protein expression were observed, as 4.7%, 5.1%, 14.0%, 13.5%, 21.8%, 19.3%, and 38.0% of non-DMD dystrophin expression levels were seen after treatment with 0.01, 0.03, 0.1, 0.3, 1, 3, and 10 μmol/L concentrations of NS-089/NCNP-02, respectively [70]. A 7-day incubation study was also conducted in DMD patient-derived myotubes. The results showed that the half-maximal effective concentration, or EC50, of NS-089/NCNP-02 was 0.63 μmol/L [70]. As PMOs have a relatively shorter half-life of 2 h in the bloodstream, this study was meant to determine whether PMOs taken up by cells in that short period could be effective and exhibit activity [71]. Exon 44 skipping activity was found to persist for at least one week following NS-089/NCNP-02 treatment [70]. Results after 4 weeks of weekly intravenous administrations to cynomolgus monkeys also showed potential, with exon-skipping observed at doses of 66, 220, and 660 mg/kg/week [66,70]. When treated for 13 weeks instead, exon skipping was observed in the cardiac muscle at doses of 600 mg/kg or higher [70]. Exon-skipping efficiencies of up to 16.2% were observed in skeletal muscle at the highest dose of 2000 mg/kg [70]. While exon skipping was observed in the myocardium, these differences in the monkey myocardial muscle were not significant, with the highest dose resulting in a 0.8% skipping efficiency. These in vitro and in vivo results led to NS Pharma initiating a phase I/II study of NS-089/NCNP-02.

### 3.2. Clinical Findings and Pharmacology

Three clinical trials have been registered for NS-089/NCNP-02 (Table 1). Results from the first clinical trial were published in 2025 [66]. Lasting from 2019 to 2022, six ambulant DMD patients between the ages of 8 and 17 years with exon 44 deletion amenable to exon skipping were intravenously administered NS-089/NCNP-02 weekly for two weeks at one of four different doses, ranging from 1.62 to 80 mg/kg/week (NCT04129294). This study was meant to assess the safety, pharmacokinetics, and efficacy of the exon-skipping treatment. This therapy was generally well tolerated across all dose levels (1.62, 10, 40, and 80 mg/kg/week) in the first dose escalation half of the study. Biceps muscle biopsies were performed pre-treatment and after the 24-week treatment period to measure dystrophin levels using western blots and immunohistochemistry [66]. Mass spectrometry and liquid chromatography were used to assess pharmacokinetics in blood samples collected during multiple timepoints pre-treatment and post-treatment on day 1, as well as on day 2, day 8, day 9, day 15, and day 22, during the first part of the study. During the second part of the study, blood samples were collected at multiple time points on day 1 before and after administration, along with collections on day 2, day 8, day 22, day 50, day 78, day 106, day 134, day 162, day 163, and either day 169 or day 176 [72]. Urinalysis used 24 h pooled urine on days 2, 4, and 11 for the first part of the study, and on days 2 and 163 for the second part of the study [72]. This therapy was generally well tolerated across all dose levels (1.62, 10, 40, and 80 mg/kg/week) in the first dose escalation half of the study. No dose-limiting toxicities or anti-dystrophin antibodies were observed [66]. The most frequent adverse events were increased β2 microglobulin levels, albumin/creatinine ratios, and cystatin C levels in the urine [66]. Other side effects at higher doses (40 or 80 mg/kg/week) included hordeolum, rash erythematous, urticaria, headache, glucose in urine, and protein in urine [66]. There was an absence of noticeable kidney toxicity at the doses tested, indicating a similar safety profile to other PMOs [66]. Overall, the absence of serious adverse events or adverse event-related discontinuations supports a tolerable safety profile at the tested dose range [66].

The secondary endpoint of this study was efficacy, which was measured through functional assessments and by examining biopsied muscle for dystrophin protein expression levels. Creatine kinase levels in the serum decreased over the 24-week treatment period in participants who received the higher 40 and 80 mg/kg/week doses. Motor functional performance was measured using the North Star Ambulatory Assessment, the Time to Run/Walk 10 Meters, the Time to Stand Test, the 6 Minute and 2 Minute Walk Tests, the Time to Climb 4 Stairs test, the Performance of Upper Limb test, grip/pinch strength tests, and muscle strength assessments (shoulder abduction, knee flexion/extension, and elbow flexion/extension) [66]. Functional performance declined in the two lower-dose cohorts (1.62 and 10 mg/kg/week) and stabilized with a trend toward improvement in the higher-dose cohorts (40 to 80 mg/kg/week) [66]. Since DMD is a progressive condition, patients between the ages of 8 and 17 typically experience a steady decline in physical function based on natural history studies [73]. Hence, stabilized performance on the above motor function tests indicates that there is a possibility for this treatment to stabilize muscle function in the long term, indicating a need for longer-term clinical studies with this investigational therapy.

Based on non-clinical and in silico analyses, there is potential for NS-089/NCNP-02 to bind to RNA products from genes other than the *DMD* gene, which results in what are known as off-target effects. These analyses found that NS-089/NCNP-02 could potentially bind to exon 8 from the phosphodiesterase 3B gene to produce a non-native, potentially immunogenic protein that could result in inflammation and fever [66]. NS-089/NCP-02 could also bind to RNA from the transient receptor potential cation channel subfamily M member 3, which could affect blood pressure [66]. Although these analyses suggested possible off-target interactions, no corresponding adverse events were observed during the trial. No cases of fever were observed, and, overall, inflammatory markers remained similar to those before treatment for most patients [66]. However, IL-6 levels did increase in week 12 and then decrease in week 24 for one participant, and a similar increase and decrease in TNF was present in another [66]. IFN-γ levels followed the same pattern in a third participant [66]. These are inflammatory markers that suggest mild inflammation, but it is important to note that DMD patients are often in a chronically inflamed state because of the constant muscle degeneration [74]. The inflammation can both stimulate regeneration and worsen muscle damage [74]. Changes in blood pressure were not reported clinically at the time of writing, but pre-clinical monkey studies observed no effect on blood pressure [66].

In the initial clinical trial for NS-089/NCNP-02, exon 44 skipping efficiency was measured before and after treatment with NS-089/NCNP-02. As previously mentioned, this clinical trial assessed both clinical outcomes and biopsy dystrophin protein levels over a 24-week period and involved six DMD patients (8–17 years of age), who received the drug via IV injection (Table 1). Western blotting and immunohistochemistry were used to assay dystrophin levels from biceps brachii muscle biopsies performed pre-treatment and after 24 weeks of treatment [66]. The results showed dose-dependent dystrophin increases of 16.63% and 24.47% after 24 weeks of treatment with 40 mg/kg and 80 mg/kg doses of NS-089/NCNP-02 [66]. The high dose (80 mg/kg) was the most effective at restoring dystrophin levels after 24 weeks. Serum proteomic data showed increases in dystrophin at 12 weeks and 24 weeks. Plasma and urinary NS-089/NCNP-02 concentrations were also measured to evaluate efficacy and used to calculate pharmacokinetic parameters, such as the urinary excretion rate [66]. An extension study was conducted following the initial Phase I/II trial and is set to be completed in 2026 (NCT05135663). Given that 20% of dystrophin restoration has been theorized to be sufficient to reduce the most severe DMD symptoms, restoration to around 25% of healthy control levels of dystrophin is promising [13]. This level of expression suggests that there is potential for long-term functional improvements, although longer-term studies are needed to confirm this finding.

Currently, NS Pharma is recruiting for a multi-center open-label Phase 2 study of NS-089/NCNP-02 (NCT05996003). This study primarily aims to assess the safety, tolerability, and pharmacokinetics of the treatment, with secondary outcome measures related to the efficacy of the treatment (Table 1). The primary outcome measures will be the number of adverse events, the pharmacokinetic parameters in urine and blood plasma, and the immunoblot results to observe the change in skeletal muscle dystrophin levels between baseline and week 25 [32]. Secondary outcome measures will use mass spectrometry, immunofluorescence staining, grip/pinch strength measures, the North Star Ambulatory Assessment, the Time to Run/Walk 10 Meters, the Time to Stand, the Total Distance of the 6-Minute Walk Test, the Time to Climb 4 Stairs, and the Performance of Upper Limb tests [32]. It will be divided into two parts with two cohorts. Participants will be ambulant, aged between 4 and 15 years, have confirmed mutations that are amenable to exon 44-skipping, and be on stable doses of glucocorticoids. They will each be intravenously administered a dose of NS-089/NCNP-02 weekly. The first cohort will undergo a 4-week treatment plan at one of three different dose levels; then, all participants will transition to a single dose level over a 24-week period. The single dose level will be the maximum tolerated dose (MTD) with acceptable side effects. Cohort 2 will undergo a 24-week treatment period at the MTD determined in the first part of the study. Patients with cardiomyopathy symptoms, or those who have received gene therapy or other ASO treatments, are excluded. This interventional study aims to conclude near the end of 2025.

Overall, the completed trials and ongoing clinical trials for NS-089/NCNP-02 have indicated a tolerable safety profile and provide dystrophin restoration levels at higher doses. Although functional stabilization was observed in the completed clinical trial, long-term efficacy and pharmacokinetics remain an area for future study. Off-target risks also warrant the continued monitoring of blood pressure and serum measurements. The completed NS-089/NCNP-02 Phase I/II trial also had a small population under study, and future trials should test the therapy in a larger cohort of DMD patients. If the current upcoming clinical trial shows a favorable safety profile and continued efficacy and functional stabilization in participants, NS-089/NCNP-02 could become a key candidate for exon 44-skipping therapy in DMD.

## 4. Comparison of Brogidirsen to Other Exon 44-Targeting Therapies

Both AOC1044 and ENTR-601-44 have conjugations that aim to improve the efficacy of the attached PMOs by enhancing bioavailability or cellular uptake. Similar to ENTR-601-44, PGN-EDO44 uses PepGen’s enhanced delivery oligonucleotide cell-penetrating peptide platform [75]. This peptide is meant to increase cellular delivery and uptake by tissues and is used in several therapeutic ASOs under development by PepGen [76]. Preclinically, mean exon 44-skipping levels of 93.4% were observed after treatment with PGN-EDO44 at the highest dose level in wild-type human myoblasts [75]. PepGen intends to continue the development of PGN-EDO44, alongside its other exon-skipping candidates. Another promising cell-penetrating peptide and PMO conjugation is DG9-PMO, which has induced 41% dystrophin expression in cardiac tissues [77]. These conjugations show a trend toward next-generation exon-skipping therapies that combine optimized oligonucleotide chemistry with advanced delivery systems, aiming to improve outcomes in DMD patients through improved pharmacokinetic profiles with enhanced tissue targeting and uptake.

With this trend in context, it is notable that NS-089 contains no conjugations with any peptides and still achieves a relatively high percentage of dystrophin restoration of around 25% in human DMD patients. Because it lacks peptide conjugations, NS-089/NCNP-02 relies on natural biodistribution and receptors to encounter and enter the skeletal and cardiac muscle cells. While these exon-skipping levels are promising, whether this effect is solely due to the dual-targeting approach or other factors, such as the relatively short 24-week treatment duration, the multiple levels of dosing, and the modest number of patients, will need further study. The dual-targeting approach of NS-089/NCNP-02, using two linked sequences targeting different sites within the same exon, differs from the single-target sequences within the AOC1044 and ENTR-601-44 therapies. This is a novel approach not only in the DMD therapeutic field but also for ASO drugs generally. A previous study has evaluated the delivery of two miRNAs with different target sites simultaneously for investigational cancer therapeutics, but no studies, at the time of writing, have conjugated two different ASO target sequences together, even in other disease contexts [66,78].

Currently, NS-089/NCNP-02 has advanced further clinically than AOC1044 and ENTR-601-44 through testing the therapy in DMD patients. NS-089/NCNP-02 is set to begin a larger Phase II trial after a Phase I/II trial in six DMD patients, while both AOC1044 and ENTR-601-44 are currently recruiting for or conducting Phase I/II trials in DMD patients after testing in healthy controls (Table 2). As a result, there is more safety and tolerability data available for NS-089/NCNP-02 in DMD patients, but results from extension studies and the EXPLORE44-OLE or ELEVATE-44-102 trials will provide more safety data to compare between all three treatments (Table 2). Some peptide-conjugated PMOs, such as AVI-5038, have caused mild tubular degeneration and renal toxicity in monkeys dosed at 9 mg/kg weekly for 4 weeks, but these toxic effects appear species- and dose-dependent [79]. Since the NS-089/NCNP-02 study had a modest number of participants and occurred over a short period, results from ongoing extension studies will also provide crucial insights into the long-term safety and clinical potential of dual-targeting antisense therapies.

In comparison to the dystrophin restoration rates of the already approved viltolarsen (pretreatment 0.3% to posttreatment 5.7%), the dystrophin increases of 24.47% for NS-089/NCNP-02 and around 20% for AOC1044 are promising [59,66]. If the dystrophin produced is effective, these dystrophin expression levels could lead to significant clinical benefits. The ability of both AOC1044 and ENTR-601-44 to target cardiac muscle tissue is also promising, given the morbidity and mortality related to cardiomyopathy in DMD patients [60].

In summary, AOC1044, ENTR-601-44, and PGN-EDO44 represent a new generation of exon-skipping therapies that use antibody or peptide conjugations to enhance tissue uptake and delivery efficiency. NS-089/NCNP-02 achieves promising dystrophin restoration without such conjugations, relying instead on a novel dual-targeting approach that may offer targeting benefits. While NS-089/NCNP-02 currently leads in clinical development with encouraging safety and efficacy data, ongoing and future trials of all candidates will be essential to determine their long-term safety, cardiac targeting efficacy, and impact on muscle function.

## 5. Addressing Challenges Related to Exon-Skipping Therapies

Given the need for treatments that address the molecular basis behind DMD and the limited efficacy of existing ASO therapies, there is a need for novel DMD therapies that exhibit increased efficacy and dystrophin restoration. Currently approved ASO therapies have lower amounts of dystrophin restoration due to endosomal entrapment, rapid bloodstream clearance, and poor muscle uptake [80]. These limitations are currently being addressed through the study of antibody conjugations, such as AOC1044, as well as by developing cell-penetrating peptides, such as those in DG9-PMO, PGN-EDO044, and ENTR-601-44 [59,64,75,77]. Further research into improving ASO-based therapies with novel delivery and targeting mechanisms is needed. Even partial restoration of dystrophin levels could lead to significant improvements during functional tests [13].

Exon 44 skipping can treat up to 6–7% of DMD patients, and it is one of several exons that can be targeted to restore the *DMD* gene reading frame. Among the exon skipping therapies, exon 51 skipping has the broadest theoretical applicability, targeting up to 13% of all DMD patients [81]. However, this is a relatively small subset of the total patient population. Even after approval, each therapy can only benefit a fraction of DMD patients, requiring the development of a separate drug for each amenable mutation. ASO therapies that skip one or two exons can have the potential to treat 83% of DMD patients (79% of all patients with deletions, 73% of all patients with duplications, and 91% of all patients with small mutations), but that leaves 17% of patients with mutations in essential dystrophin protein-coding regions without ASO therapies [40]. Since exon-skipping ASO therapies need to be personalized for specific mutations, each new ASO therapy needs to invest time and funds into the regulatory process before reaching patients [82].

One option that could treat a larger group of DMD patients would be to administer multiple ASO exon-skipping therapies together as a ‘cocktail’ [83]. This approach, when applied to exons 45–55, would theoretically be applicable to up to 63–65% of DMD patients with deletion mutations, as the combination of multiple different ASOs would induce the skipping of multiple exons [37,84]. Preclinical studies have multi-exon-skipping efficiencies of up to 15% for exons 45 to 55 in a humanized DMD mouse model treated at a dose of 1.67 μg/PMO [37]. Adding dual-targeting antisense therapies into these cocktail mixtures could be an area of potential future study. However, multi-exon-skipping mixtures would require more complex approval processes under the current regulatory processes than single-exon-skipping therapies. This could incur significantly higher costs and time investments than individual ASO treatments.

Depending on which exons are skipped, different truncated dystrophins are produced, and there are functional differences between truncated in-frame dystrophins [85]. Certain deletions could result in altered interactions with other proteins and instability within the dystrophin protein itself. For example, deletions of exons 45 to 46 in frame often result in a DMD phenotype, but deletions of exons 45 to 47 or 45 to 48 often result in a milder BMD phenotype [34]. Hence, measuring efficacy using the percentage of dystrophin restoration may not necessarily result in actual functional improvements in patients. This is due to the dystrophin protein structure, with some domains being more crucial and less skippable than others for sarcolemma protein interactions [2]. It is also important to note that shorter truncated dystrophins, such as those with exon 45 to 55 deletions and exon 3 to 9 deletions, are associated with a milder phenotype than longer proteins, which adds complexity [39,86]. Some currently approved ASO therapies, such as eteplirsen and golodirsen, had controversial approvals that were based on increases in dystrophin expression without definitive conclusions about functional improvements [30,82,87]. Modest clinical benefits were observed for these treatments years after initial FDA approval, with further clinical trials (NCT03992430; NCT02500381) still ongoing [88,89,90]. Based on these approvals and the different truncated dystrophins produced, it is possible that the efficacy of current ASOs in the clinical stage may not be fully characterized for years, even if these therapies obtain regulatory body approval.

Ideally, molecular DMD therapies should restore dystrophin in the respiratory, skeletal, and cardiac muscles. With limited dystrophin increases observed in cardiac tissue after NS-089/NCNP-02 administration, other approaches are needed to target cardiac muscle. Research into alternative approaches, such as gene therapies, CRISPR-based therapies, cell-based therapies, and other mechanisms, should still be continued. Numerous clinical trials are underway to evaluate these alternative approaches; however, most strategies, aside from those repurposing existing treatments approved for other conditions, continue to face similar regulatory challenges, with unique safety and efficacy issues that contribute to a 4.6% DMD clinical trial success rate [91,92]. For example, gene therapies were initially intended to bypass the need for repeated doses of existing ASO treatments and were thought to be applicable for almost all DMD patients, but instead led to the implementation of exclusion criteria following several immune reactions against the expressed foreign micro-dystrophin proteins [53,93,94]. There is an unmet need for alternative, cost-effective, dystrophin restoration methods. Promising treatment approaches should build on existing results to predict and enhance their tolerability in a population with elevated health complication risks and care needs.

## 6. Conclusions

Overall, NS-089/NCNP-02 is a promising DMD therapeutic candidate for dystrophin restoration in skeletal muscle that uses a novel dual-targeting oligonucleotide strategy. This novel approach significantly broadens the landscape of ASO therapy design, which may improve therapeutic efficacy and applicability across a wider range of mutations. The use of a single dual-targeting sequence may provide an alternative treatment pathway that circumvents existing patent constraints and avoids reliance on antibody or peptide conjugation. When combined with antibody or peptide conjugates, this strategy could further enhance delivery efficiency and therapeutic potency. Considerable progress has been made in the application of antisense therapy for DMD, and NS-089/NCNP-02 shows promising efficacy in skeletal muscles that exceeds that of currently approved ASO therapies for DMD. Molecular strategies that are more broadly applicable to all DMD patients could significantly improve long-term outcomes and lower clinical development costs for a larger group of patients.

## Figures and Tables

**Figure 1 genes-16-00777-f001:**
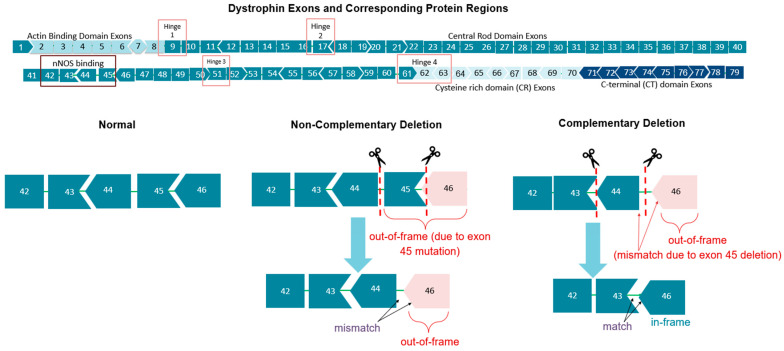
Dystrophin exons and the effect of different exon deletions on the RNA reading frame. Differently shaped exons represent which exons can ‘fit’ together to restore a functional reading frame.

**Figure 2 genes-16-00777-f002:**
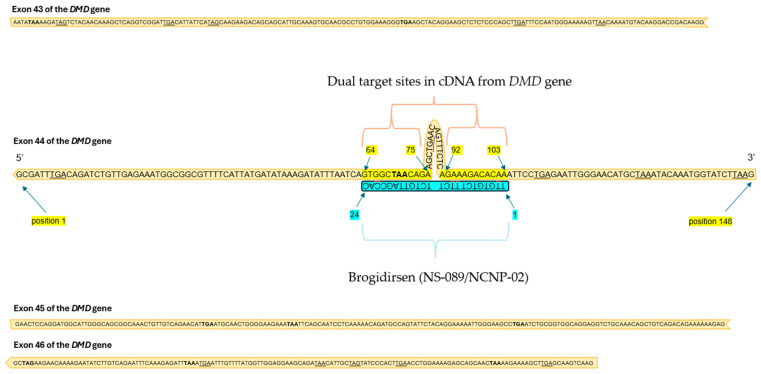
Target sites of brogidirsen (NS-089/NCNP-02) within the exon 44 sequence of the *DMD* gene. Bolding and underlining indicate stop codons.

**Figure 3 genes-16-00777-f003:**
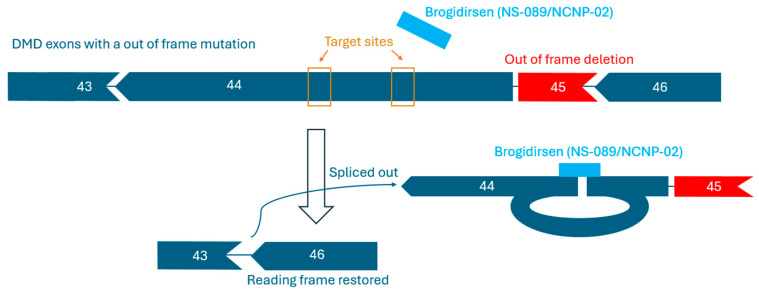
Reading frame restoration mechanism of brogidirsen. The two sites within brogidirsen both target sequences within exon 44 of the *DMD* gene.

**Table 1 genes-16-00777-t001:** Summary of criteria and comparison table for all brogidirsen clinical trials in 2025.

ClinicalTrials.gov ID	NCT04129294	NCT05135663	NCT05996003
Phase	I/II	II	II
Start date	December 2019	June 2021	February 2024
(Estimated) End date	May 2022	July 2026	November 2025
Description	Dose-escalation, open-label	Open-label, extension study	Open-label, multi-center
Dose		80 mg/kg, 40 mg/kg	
Primary Endpoint	Safety, tolerability	Safety	Safety, pharmacokinetics
Secondary Endpoints	Pharmacokinetics, efficacy	Efficacy	Efficacy
Enrollment numbers	6	6	20 (6 in Cohort 1, 14 in Cohort 2)
Route	Intravenous	Intravenous	Intravenous
Life expectancy	At least one year	Same participants as NCT04129294	N/A
Age Range	8–17	8–17	4–15
Design	24 weeks of treatment, 12-week follow-up period	Administer once weekly for 216 weeks	Once weekly for 4 weeks at 1 of 3 doses, 24 weeks at MTD ^1^ after
Corticosteroid use	None, or at least 6 months of stable use	Same participants as NCT04129294	Stable dose for at least 3 months
Ambulation Requirements	Ambulant	Same participants as NCT04129294	Able to walk independently without devices
Exclusion Criteria	No DNA polymorphisms that could compromise therapy and pre-mRNA binding, FVC ^1^ < 50% of predicted, EF ^1^ < 40%, FS ^1^ < 25% based on ECHO ^1^, current infections, cardiomyopathy, liver/renal disease, previous severe drug allergy, continuous use of artificial respirator, previous use of investigational therapies	Same participants as NCT04129294	Body weight of <20 kg, cardiomyopathy symptoms, use of anabolic steroids, use of other investigational drugs in the past three months, surgery in last 3 months, taken gene therapy or another exon skipping drug
Other Criteria	Adequate intact muscles for biopsy, able to give written informed consent, QTc ^1^ < 450 ms (<480 ms for subjects with Bundle Branch Block)	Must have participated in NCT04129294	Able to complete the TTSTAND ^1^ without assistance in <20 s,
Test Timing	At the end of the treatment period (24 weeks)	Up to Week 243	Baseline, Week 13, Week 25
Genotype	Out-of-frame deletion(s) amenable to exon 44 skipping	Out-of-frame deletion(s) amenable to exon 44 skipping	Amenable to exon 44 skipping
Reference	[66]	[72]	NCT05996003

^1^ forced vital capacity (FVC), left ventricular ejection fraction (EF), fractional shortening (FS), echocardiogram (ECHO), Corrected QT Interval (QTc).

**Table 2 genes-16-00777-t002:** Comparison of AOC1044, ENTR-601-44, and NS-089/NCNP-02 strategies and current clinical progress.

Exon-Skipping Therapies	AOC1044 (Delpacibart Zotadirsen)	ENTR-601-44	NS-089/NCNP-02 (Brogidirsen)
**Modification**	Antibody-conjugated	Peptide-conjugated	Unmodified
**Details**	4 PMOs conjugated to anti-TfR1	PMO conjugated to EEV	No conjugations
**Targeting Mechanism**	TfR1 for targeted delivery	EEV to escape endosomes	Natural biodistribution
**Target site(s)**	Single	Single	Dual
**Regulatory State**	No clinical holds	FDA hold lifted in 2025	No clinical holds
**Current Clinical Stage**	Phase II (Part B of EXPLORE44-OLE, enrolling DMD patients)	Phase Ib (ELEVATE-44-102 trial, enrolling DMD patients)	Phase II (NCT05996003, recruiting DMD patients)
**Age**	7 to 27 years old		
**Reported clinical trial design(s)**	40 healthy volunteers in a double-blind, placebo-controlled trial	4 dose levels tested in healthy males	4 dose levels tested in 6 DMD patients
	25 patients receiving AOC1044	32 patients to be enrolled	20 patients to be enrolled

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
