# Peer review of "Brogidirsen and Exon 44 Skipping for Duchenne Muscular Dystrophy: Advances and Challenges in RNA-Based Therapy"

_genes, 2025, doi:10.3390/genes16070777_

Round 1

Reviewer 1 Report

Comments and Suggestions for Authors

The review article by Tang and Yokota is a well-constructed review of the clinical product development program of Brogidirsen a dystrophin exon 44 skipping therapy that is currently moving forward in Phase 2 clinical trials at this point.  The review nicely discusses the novelty of this unmodified phosphorodiamidate morpholino oligonucleotide (PMO) targets two internal splice regulators within exon 44 and facilitates processing of the pre-mRNA to an mRNA lacking both exons 44 and 45, which restores the open reading frame of dystrophin (figure 2).  This dual-targeting design is thought to enhance the potency of brogidirsen and obviate the need for conjugation with any form of cell-penetrating peptide or antibody, as was done with two other exon 44 skipping oligos.  The review focuses on the clinical trial designs and results, tracing the progress of brogidirsen and contrasting it with that of the other two exon 44 skipping products.  

Overall, this review is well-written, clear and concise.  Reading this review would be valuable for neuromuscular neurologists and basic scientists interested in DMD product development.  In addition the dual-targeting design could potentially be applied to exon skipping for genetic disorders with gene structures similar to dystrophin.  

The manuscript in question is a review article rather than an original article, so methodologic considerations must be considered in that light.  Furthermore, it is not intended to be a comprehensive review but rather a comparison of products in development within a rather narrow context, that of oligonucleotide therapies for mutations in the human dystrophin gene.  Finally, it is comparing products that are not yet commercially approved.  In such a case, the limited amount of published data on each agent must be complemented by clinical trial information available from other sources in the public domain.

Despite the narrow scope of the paper, the comparison presented is useful because it illustrates how certain design features can influence the clinical attributes of the oligonucleotide drug.  

If one were to suggest changes in the manuscript to make it more useful, the manuscript could include that statement  explicitly in the manuscript As follows.. 

"Despite the narrow scope of this paper, the comparison presented is useful because it illustrates how certain design features can influence the clinical attributes of the oligonucleotide drug.  

The design of a single linear PMO sequence targeting two discrete internal splice regulators with exon 44 appears to add sufficient potency to offset any potential need for conjugation to a targeting antibody or cell-penetrating peptide, as is done with the other exon 44 skipping oligos in clinical development.

Author Response

Reviewer Comment 1 - please also see attached pdf copy, to preserve the Table formatting

This paper reviews the current situation regarding trials of Exon 44 ASOs for potential therapy in DMD patients who have an out-of-frame deletion in the dystrophin gene which can be converted to (effectively) a larger in-frame deletion through ASO-mediated splicing-out of Exon 44.

This is an exciting and promising field for therapy in some specific patients with DMD, and this current review of the progress is helpful.

However, some of the essential detail of the background information which is required for a non-specialist to be able to appreciate the approach to design of treatments, and their mechanism of action, has been given only cursory explanation, through perhaps overestimating the background familiarity of potential readers.

Response 1: Thank you to the reviewer for your efforts and feedback during the review process (the table is particularly helpful). We have done our best to incorporate this reviewer’s feedback into our manuscript.

Major Point
Specifically, the introduction needs to discuss the exon structure of the dystrophin gene in relation to each exon being one of 3n, 3n+1 or 3n-1 nucleotides so that exon deletions, as the most common type of dystrophin mutation, are of 3 types in relation to their effect on the reading frame. An additional introduced ‘deletion’ must complement an existing out-of-frame deletion if it is to correct the reading frame. If Exon 44 is 178 nucleotides (as in Figure 1), it would be a 3n+1 type, and its induced deletion can only compensate for deletions which are of 3n-1 type. This can be best achieved if the authors could include a more general diagram of the dystrophin gene exons, together with a simple explanation, similar to that used by Muscular Dystrophy UK to explain exon skipping in lay terms (eg.  see https:www.musculardystrophyuk.org/research/what-is-exon-skipping-and-how-does-it-work/

Response 2: Thank you for bringing up this point. This information has been added in (see the 3rd paragraph in section 1.2), along with a new figure (Figure 1).

Minor Points
There are several other minor points which require attention, and are listed in the Table below (together with additional discussion of the point above)

N

Page/
line

Current text

Suggested revised text

Comments

1

P2,L43-44 & 48-49

…death from respiratory or cardiac complications in the thirties to forties [7,8].

…AND…

Current palliative therapies have extended the lifespan of some DMD patients from around 20 years to 30 - 40 years [10].

Response 3: Thank you for the comment. The text has been revised accordingly with additional information.

These two statements, as written, seem inconsistent if the first sentence is describing the untreated natural history.

Please review and amend /clarify in the text, accordingly.

2

P2, L55

(and similarly P4,
L159)

CRISPR-Cas9,

Response 4: Thank you for this comment. The words “delivery systems” have been added in.

CRISPR-Cas9 is a vehicle for treatment, rather than a description of the engineered therapeutic alteration of the DNA or RNA.   Please use a term here which describes what is happening, to fit in with the other two terms of gene-delivery, and  exon skipping.

3

P2. L61

3 to 4

Response 5: This change has been implemented.

3 to 4 years

4

P2, L63

age 5

Response 6: This change has been added in.

age 5 years

5

P2, L64

…gene testing, creatine kinase tests…

Response 7: This change has been put in.

…creatine kinase tests, gene testing…

It makes more sense to put these in chronological order, given the next sentence:  ‘.. creatine kinase levels are typically tested early, and genetic tests are then used to confirm the diagnosis..’

6

P2, L66-67

.. number of mutations..

Response 8: This has been added in as ‘pathogenicity scale numbering’ instead of ‘number’.

…exact descriptive notation for genetic alterations or variants…

OR

… the pathogenicity-scale numbering of all variants..… 

Please find an appropriate form of words here

If talking about the ‘number’ of ‘mutations’, some of these are likely to be sequence variants which may or may not be significant.  Multiple pathogenic variants would be very rare, except perhaps in very occasional fully-manifesting DMD females who could (theoretically) have an affected (BMD) father plus a de-novo 2nd pathogenic variant.

7

P2, L70-71

Interestingly, female carriers of DMD can exhibit elevated creatine kinase as well [18].

Response 9: This change has been added in.

Interestingly, female carriers of DMD can exhibit elevated creatine kinase as well [18], usually as a shifted distribution of levels compared with the ‘normal range’, though can be markedly raised in some clinically symptomatic manifesting carriers. 

The degree of elevation of CK in most carriers is typically still 40-50x less than the levels seen in DMD boys. This sentence therefore needs qualifying so that it does not imply that carriers can be expected to have CK levels equivalent to those in affected males.   Of course, very occasionally they can, and fully manifest for DMD clinically (eg. in 45,XO females who happen to have a pathogenic DMD mutation on their single X chromosome).

8

P3, L104-115

Whole Paragraph

Response 10: This change has been added in.

It is important in this paragraph to introduce also the concept of in-frame and out-of frame DNA pathogenic mutations, and that exon- skipping may require more than one exon to be skipped in order to return to in-frame reading of the transcribed RNA.

9

P3, L125

… due othe…

Response 11: This change has been implemented. Thank you for noticing.

…due to the…

Typo

10

P3, L130-2

ASOs can be structurally classified into Locked Nucleic Acid (LNA), 2′-O-methyl RNA backbone (MOE), and phosphorodiamidate morpholino oligomer (PMO) subclass groupings [40].

Response 12: Thank you for this comment. More information about LNA and MOE molecules have been added in.

The authors describe PMOs in the next sentences. They need here to give a few words to describe what LNA and MOE are, so that the difference between these and PMO is clear.

11

P4, L147

… result in little interactions with biological molecules…

Response 13: This change has been added in, thank you for noticing.

???  … result in little interaction with biological molecules… ???

Is this a simple Typo (‘interaction’ is better than ‘interactions’), or do the authors really mean ‘there are multiple little interactions with biological molecules’ – if so these little interactions need further description.

12

P4, L159

CRISPR-Cas-9

Response 14: This has been added in with a similar modification (‘delivery mechanisms’) as  for comment 2.

See Comment No 2 above

13

P4, L176-178

skipping exon 44 is thought to result in the restoration of the reading frame and yield a shortened but partially functional dystrophin protein [33].

Response 15: The bolded text has been added in.

skipping exon 44 in DMD boys who already have an exon 45 deletion is thought to result in the restoration of the reading frame and yield a shortened but partially functional dystrophin protein [33].

The authors need to make it clear that it is not skipping exon 44 per se that gives a milder phenotype – indeed on its own it does not, but in combination with certain other neighbouring out-of-frame deletions.  A good simple diagram for this can be found at:

https:www.musculardystrophyuk.org/research/what-is-exon-skipping-and-how-does-it-work/

The authors must in the present paper give a clear description of the difference between exons with 3n bases and exons with 3n +/- 1 bases, as the whole basis for exon 44 deletion as a potential therapy in DMD with neighbouring exon deletions, as this is fundamental to understanding this paper. 

14

P5, L 200-201

In preclinical models,

Response 16: This information has been added in.

In preclinical models with exon 45 deletion ?

Preclinical models with what mutations ?

15

P5,L205-208

The safety and tolerability of this therapy was tested in a placebo-controlled, randomized, double-blind EXPLORE44 Phase I/II clinical trial on 40 healthy volunteers (NCT05670730). Results showed the AOC1044 treated group only  had mild or moderate adverse events, indicating the therapy was well tolerated [53].

Response 17: Authors agree with this reviewer’s comments and can confirm it was really given to healthy male human controls, but cannot find applicable literature for why there were no serious effects. It may be because of the low amount of dystrophin protein needed to maintain function, and also the time frame (one dose, then monitored for 3 months).

Was an Exon 44 ASO really given to healthy male human controls – when exon 44 is an out-of-frame exon ? – or is it ok because it blocks exon 44 in only a small proportion of muscle cells ?  This needs explanation in the text as to how it can have no effect if given alone.

16

P5,L238-240

…in a previous trial by Nippon-Shinyaku and NS Pharma (NCT04129294), w… where treatment was shown to restore dystrophin expression to approximately 25% of the levels observed in healthy controls [63,65]…

Response 18: This sentence has been rewritten into: “. It has shown therapeutic potential in a previous preclinical study, where pa-tient-derived MYOD-converted fibroblasts with DMD exon 45 deletion resulted in a EC50 of 0.63 μmol/L for exon 44 skipping 7 days after a 1 hour incubation with the NS-089/NCNP-02 treatment [66]. In a following human clinical trial by Nip-pon-Shinyaku and NS Pharma (NCT04129294), treatment was shown to restore dys-trophin expression to approximately 25% of the levels observed in healthy controls [63].”

If these are pre-clinical studies, the authors here must say on what biological sample it was run – ie. was it on cultured myoblasts, or on an animal model ? 

Also, the Nippon-Shinyaku trial is reference 64 (not 65) and the figures quoted in that 2023 trial are for 17% normal levels.  However the  figures in Reference 63, which appears to be a clinical DMD patient trial (rather than pre-clinical)  are indeed close to 25% restoration.

The authors need to rewrite this sentence accordingly.

17

P7,L278-281

Figure 1.

Response 19: Exon 46 has been changed to blue. The authors intended for the red to indicate out of frame and not a deletion, but understand it can be confusing and changed both the color and wording.

Exon 46 should be marked in blue, not in red ?  Also the ‘Out of frame’ in red should be over Exon 45, not exons 45 and 46 ?

Surely, this diagram should be of treating an Exon 45 deletion (Out of frame), as Exon 45-46 deletion would appear to be in-frame.

18

P7, 288-289

with exon 44 deletion amenable to exon skipping

Response 20: Thank you for noticing this error. The manuscript has been reworded to “exon deletions amenable to exon 44 skipping”.

with exon 45 deletion amenable to exon skipping

Typo ?  Surely the authors mean Exon 45 deletion.

19

P7, L 291-292

…efficacy of the exon skipping treatment.  This therapy was generally well tolerated…

Response 21: Three sentences have been added in between these two to better cover the methodology.

Between these two sentences the authors must indicate some key methodology description – particularly relating to the frequency of clinical assessments, the timing of muscle biopsies, and the site of those biopsies, and the method of dystrophin assay.

See Point 20 below

20

P8L338-P9L350

Clinical trial of NS-089/NCNP-02

Response 22: A sentence covering the methods has been added in, along with another that restates trial details in the paragraph.

The authors must indicate here how often were the muscle biopsies done, which muscles were biopsied, and how were dystrophin levels assayed.

..Also, some of the salient points in the paragraphs l286-300 and L302-317 should be repeated here. Ie. It should restated here that it was a clinical trial of 6 DMD patients (8-17 years) receiving the drug by IV injection, and assessing both clinical outcome and biopsy dystrophin level over a 24-week treatment period.

21

P10, L400-401

Table 1

Response 23: The table now has rows, thank you for pointing that out.

In the pdf provided for peer review, there is no delineation of the different rows in Table 1, which unfortunately run into each other. I suspect this is merely a reproductional formatting issue, but must be corrected for a published version.

Reviewer 2 Report

Comments and Suggestions for Authors

DMD: Exon 44 skipping by Brogidirsen   Genes 3693221

Reviewer Comments - please also see attached pdf copy, to preserve the Table formatting 

This paper reviews the current situation regarding trials of Exon 44 ASOs for potential therapy in DMD patients who have an out-of-frame deletion in the dystrophin gene which can be converted to (effectively) a larger in-frame deletion through ASO-mediated splicing-out of Exon 44.

This is an exciting and promising field for therapy in some specific patients with DMD, and this current review of the progress is helpful.

However, some of the essential detail of the background information which is required for a non-specialist to be able to appreciate the approach to design of treatments, and their mechanism of action, has been given only cursory explanation, through perhaps overestimating the background familiarity of potential readers.

Major Point
Specifically, the introduction needs to discuss the exon structure of the dystrophin gene in relation to each exon being one of 3n, 3n+1 or 3n-1 nucleotides so that exon deletions, as the most common type of dystrophin mutation, are of 3 types in relation to their effect on the reading frame. An additional introduced ‘deletion’ must complement an existing out-of-frame deletion if it is to correct the reading frame. If Exon 44 is 178 nucleotides (as in Figure 1), it would be a 3n+1 type, and its induced deletion can only compensate for deletions which are of 3n-1 type. This can be best achieved if the authors could include a more general diagram of the dystrophin gene exons, together with a simple explanation, similar to that used by Muscular Dystrophy UK to explain exon skipping in lay terms (eg.  see https:www.musculardystrophyuk.org/research/what-is-exon-skipping-and-how-does-it-work/

Minor Points
There are several other minor points which require attention, and are listed in the Table below (together with additional discussion of the point above)

N

Page/
line

Current text

Suggested revised text

Comments

1

P2,L43-44 & 48-49

…death from respiratory or cardiac complications in the thirties to forties [7,8].

…AND…

Current palliative therapies have extended the lifespan of some DMD patients from around 20 years to 30 - 40 years [10].

These two statements, as written, seem inconsistent if the first sentence is describing the untreated natural history.

Please review and amend /clarify in the text, accordingly.

2

P2, L55

(and similarly P4,
L159)

CRISPR-Cas9,

CRISPR-Cas9 is a vehicle for treatment, rather than a description of the engineered therapeutic alteration of the DNA or RNA.   Please use a term here which describes what is happening, to fit in with the other two terms of gene-delivery, and  exon skipping.

3

P2. L61

3 to 4

3 to 4 years

4

P2, L63

age 5

age 5 years

5

P2, L64

…gene testing, creatine kinase tests…

…creatine kinase tests, gene testing…

It makes more sense to put these in chronological order, given the next sentence:  ‘.. creatine kinase levels are typically tested early, and genetic tests are then used to confirm the diagnosis..’

6

P2, L66-67

.. number of mutations..

…exact descriptive notation for genetic alterations or variants…

OR

… the pathogenicity-scale numbering of all variants..…  

Please find an appropriate form of words here

If talking about the ‘number’ of ‘mutations’, some of these are likely to be sequence variants which may or may not be significant.  Multiple pathogenic variants would be very rare, except perhaps in very occasional fully-manifesting DMD females who could (theoretically) have an affected (BMD) father plus a de-novo 2nd pathogenic variant.

7

P2, L70-71

Interestingly, female carriers of DMD can exhibit elevated creatine kinase as well [18].

Interestingly, female carriers of DMD can exhibit elevated creatine kinase as well [18], usually as a shifted distribution of levels compared with the ‘normal range’, though can be markedly raised in some clinically symptomatic manifesting carriers.  

The degree of elevation of CK in most carriers is typically still 40-50x less than the levels seen in DMD boys. This sentence therefore needs qualifying so that it does not imply that carriers can be expected to have CK levels equivalent to those in affected males.   Of course, very occasionally they can, and fully manifest for DMD clinically (eg. in 45,XO females who happen to have a pathogenic DMD mutation on their single X chromosome).

8

P3, L104-115

Whole Paragraph

It is important in this paragraph to introduce also the concept of in-frame and out-of frame DNA pathogenic mutations, and that exon- skipping may require more than one exon to be skipped in order to return to in-frame reading of the transcribed RNA.

9

P3, L125

… due othe…

…due to the…

Typo

10

P3, L130-2

ASOs can be structurally classified into Locked Nucleic Acid (LNA), 2′-O-methyl RNA backbone (MOE), and phosphorodiamidate morpholino oligomer (PMO) subclass groupings [40].

The authors describe PMOs in the next sentences. They need here to give a few words to describe what LNA and MOE are, so that the difference between these and PMO is clear.

11

P4, L147

… result in little interactions with biological molecules…

???  … result in little interaction with biological molecules… ???

Is this a simple Typo (‘interaction’ is better than ‘interactions’), or do the authors really mean ‘there are multiple little interactions with biological molecules’ – if so these little interactions need further description.

12

P4, L159

CRISPR-Cas-9

See Comment No 2 above

13

P4, L176-178

skipping exon 44 is thought to result in the restoration of the reading frame and yield a shortened but partially functional dystrophin protein [33].

skipping exon 44 in DMD boys who already have an exon 45 deletion is thought to result in the restoration of the reading frame and yield a shortened but partially functional dystrophin protein [33].

The authors need to make it clear that it is not skipping exon 44 per se that gives a milder phenotype – indeed on its own it does not, but in combination with certain other neighbouring out-of-frame deletions.  A good simple diagram for this can be found at:

https:www.musculardystrophyuk.org/research/what-is-exon-skipping-and-how-does-it-work/

The authors must in the present paper give a clear description of the difference between exons with 3n bases and exons with 3n +/- 1 bases, as the whole basis for exon 44 deletion as a potential therapy in DMD with neighbouring exon deletions, as this is fundamental to understanding this paper.  

14

P5, L 200-201

In preclinical models,

In preclinical models with exon 45 deletion ?

Preclinical models with what mutations ?

15

P5,L205-208

The safety and tolerability of this therapy was tested in a placebo-controlled, randomized, double-blind EXPLORE44 Phase I/II clinical trial on 40 healthy volunteers (NCT05670730). Results showed the AOC1044 treated group only  had mild or moderate adverse events, indicating the therapy was well tolerated [53].

Was an Exon 44 ASO really given to healthy male human controls – when exon 44 is an out-of-frame exon ? – or is it ok because it blocks exon 44 in only a small proportion of muscle cells ?  This needs explanation in the text as to how it can have no effect if given alone.

16

P5,L238-240

…in a previous trial by Nippon-Shinyaku and NS Pharma (NCT04129294), w… where treatment was shown to restore dystrophin expression to approximately 25% of the levels observed in healthy controls [63,65]…

If these are pre-clinical studies, the authors here must say on what biological sample it was run – ie. was it on cultured myoblasts, or on an animal model ?  

Also, the Nippon-Shinyaku trial is reference 64 (not 65) and the figures quoted in that 2023 trial are for 17% normal levels.  However the  figures in Reference 63, which appears to be a clinical DMD patient trial (rather than pre-clinical)  are indeed close to 25% restoration.

The authors need to rewrite this sentence accordingly.

17

P7,L278-281

Figure 1.

Exon 46 should be marked in blue, not in red ?  Also the ‘Out of frame’ in red should be over Exon 45, not exons 45 and 46 ?

Surely, this diagram should be of treating an Exon 45 deletion (Out of frame), as Exon 45-46 deletion would appear to be in-frame.

18

P7, 288-289

with exon 44 deletion amenable to exon skipping

with exon 45 deletion amenable to exon skipping

Typo ?  Surely the authors mean Exon 45 deletion.

19

P7, L 291-292

…efficacy of the exon skipping treatment.  This therapy was generally well tolerated…

Between these two sentences the authors must indicate some key methodology description – particularly relating to the frequency of clinical assessments, the timing of muscle biopsies, and the site of those biopsies, and the method of dystrophin assay.

See Point 20 below

20

P8L338-P9L350

Clinical trial of NS-089/NCNP-02

The authors must indicate here how often were the muscle biopsies done, which muscles were biopsied, and how were dystrophin levels assayed.

..Also, some of the salient points in the paragraphs l286-300 and L302-317 should be repeated here. Ie. It should restated here that it was a clinical trial of 6 DMD patients (8-17 years) receiving the drug by IV injection, and assessing both clinical outcome and biopsy dystrophin level over a 24-week treatment period.

21

P10, L400-401

Table 1

In the pdf provided for peer review, there is no delineation of the different rows in Table 1, which unfortunately run into each other. I suspect this is merely a reproductional formatting issue, but must be corrected for a published version.

Author Response

The review article by Tang and Yokota is a well-constructed review of the clinical product development program of Brogidirsen a dystrophin exon 44 skipping therapy that is currently moving forward in Phase 2 clinical trials at this point.  The review nicely discusses the novelty of this unmodified phosphorodiamidate morpholino oligonucleotide (PMO) targets two internal splice regulators within exon 44 and facilitates processing of the pre-mRNA to an mRNA lacking both exons 44 and 45, which restores the open reading frame of dystrophin (figure 2).  This dual-targeting design is thought to enhance the potency of brogidirsen and obviate the need for conjugation with any form of cell-penetrating peptide or antibody, as was done with two other exon 44 skipping oligos.  The review focuses on the clinical trial designs and results, tracing the progress of brogidirsen and contrasting it with that of the other two exon 44 skipping products. 

Overall, this review is well-written, clear and concise.  Reading this review would be valuable for neuromuscular neurologists and basic scientists interested in DMD product development.  In addition the dual-targeting design could potentially be applied to exon skipping for genetic disorders with gene structures similar to dystrophin. 

The manuscript in question is a review article rather than an original article, so methodologic considerations must be considered in that light.  Furthermore, it is not intended to be a comprehensive review but rather a comparison of products in development within a rather narrow context, that of oligonucleotide therapies for mutations in the human dystrophin gene.  Finally, it is comparing products that are not yet commercially approved.  In such a case, the limited amount of published data on each agent must be complemented by clinical trial information available from other sources in the public domain.Despite the narrow scope of the paper, the comparison presented is useful because it illustrates how certain design features can influence the clinical attributes of the oligonucleotide drug. 

Response 1: Thank you for taking the time to review our manuscript. We greatly appreciate your feedback.

If one were to suggest changes in the manuscript to make it more useful, the manuscript could include that statement  explicitly in the manuscript As follows..

"Despite the narrow scope of this paper, the comparison presented is useful because it illustrates how certain design features can influence the clinical attributes of the oligonucleotide drug.  The design of a single linear PMO sequence targeting two discrete internal splice regulators with exon 44 appears to add sufficient potency to offset any potential need for conjugation to a targeting antibody or cell-penetrating peptide, as is done with the other exon 44 skipping oligos in clinical development.

Response 2: These sentences have been added at the bottom of section 1.2.

Reviewer 3 Report

Comments and Suggestions for Authors

The manuscript addresses an important and timely topic—the therapeutic potential of exon 44 skipping in Duchenne muscular dystrophy, with a focus on brogidirsen and NS-089/NCNP-02. The subject is relevant, and the intention to compare available therapies is appreciated. However, several aspects limit the manuscript's overall impact.

General remarks:

  • the sections provided in the manuscript indicate a numbering issue or a missing Section 3. Specifically, it jumps from Section 2 to Section 4. Moreover, there are two sections both labeled as Section 6.
  • This suggests either the omission of Section 3 or a misnumbering that should be addressed to ensure proper organization of the manuscript.
  • The Sections are:”1. Introduction to Duchenne Muscular Dystrophy and Treatments; 2. Exon 44 Skipping Therapies; 4. Brogidirsen studies and trials; 5. Comparison of Brogidirsen to other Exon 44 Targeting Therapies; 6. Addressing Challenges with Exon Skipping Therapies; 6. Conclusion. ”Comments to the Authors:

    1. Introduction Section:
    • The authors should clarify the purpose of this review in the Introduction. The current narrative does not make it clear that the article is centred on Brogidirsen and its comparison to other exon 44 skipping approaches.
    1. Exon 44 Skipping Therapies:
    • How do clinicians identify patients who are candidates? Genotyping is implied but not discussed.
    • Each therapy is described in isolation. A short sentence comparing their stages or platforms would help the reader understand where each stands in the pipeline.
    • There is no concluding synthesis that summarises the current state of exon 44 skipping therapy development and what the next steps are.
    1. Section 3?
    2. Brogidirsen studies and trials:
    • Are Figures 1 and 2 the authors’ original work, or have they been adapted from previously published sources? If adapted, have the appropriate permissions been obtained and cited accordingly?
    • The information presented is more descriptive. It would help to include a more critical discussion of trial design, endpoints, and potential limitations.

    6.Conclusion:

    - The conclusion should not only summarise the findings but also clearly state the implications for future research, clinical practice, or policy.

Comments on the Quality of English Language

I am not in a position to formally assess the quality of English. However, the text is generally understandable and does not hinder comprehension.

Author Response

The manuscript addresses an important and timely topic—the therapeutic potential of exon 44 skipping in Duchenne muscular dystrophy, with a focus on brogidirsen and NS-089/NCNP-02. The subject is relevant, and the intention to compare available therapies is appreciated. However, several aspects limit the manuscript's overall impact.

Response 1: We thank the reviewer for their comments and time spent reviewing. We have carefully considered their suggestions and have incorporated revisions accordingly.

General remarks:

  • the sections provided in the manuscript indicate a numbering issue or a missing Section 3. Specifically, it jumps from Section 2 to Section 4. Moreover, there are two sections both labeled as Section 6.
  • This suggests either the omission of Section 3 or a misnumbering that should be addressed to ensure proper organization of the manuscript.
  • The Sections are:”1. Introduction to Duchenne Muscular Dystrophy and Treatments; 2. Exon 44 Skipping Therapies; 4. Brogidirsen studies and trials; 5. Comparison of Brogidirsen to other Exon 44 Targeting Therapies; 6. Addressing Challenges with Exon Skipping Therapies; 6. Conclusion.
  • Response 2: Thank you to the reviewer for noticing. This was a formatting issue that has been fixed.
  • ”Comments to the Authors:
    • Introduction Section:
    • The authors should clarify the purpose of this review in the Introduction. The current narrative does not make it clear that the article is centred on Brogidirsen and its comparison to other exon 44 skipping approaches.

Response 3: A paragraph has been added in at the end of section 1.2 to address this comment.

  • Exon 44 Skipping Therapies:
  • How do clinicians identify patients who are candidates? Genotyping is implied but not discussed.

Response 4: Thank you for this comment. The sentence “clinicians identify patient candidates who are amenable for exon 44 skipping through genotyping” has been added in to address this comment.

  • Each therapy is described in isolation. A short sentence comparing their stages or platforms would help the reader understand where each stands in the pipeline.
  • There is no concluding synthesis that summarises the current state of exon 44 skipping therapy development and what the next steps are.

Response 5: This is a good point, but the information comparing where the trials stand is in the 3rd paragraph of section 4 and concluding information about exon 44 skipping next steps is addressed in section 5. However, a sentence has been added in the last paragraph of section 2.2 as a quick comparison.

  • Section 3?

Response 6: This has been addressed.

  • Brogidirsen studies and trials:
  • Are Figures 1 and 2 the authors’ original work, or have they been adapted from previously published sources? If adapted, have the appropriate permissions been obtained and cited accordingly?

Response 7: The figures are original ones, made based on information provided from articles cited in the text.

  • The information presented is more descriptive. It would help to include a more critical discussion of trial design, endpoints, and potential limitations.

Response 8: Thank you for this comment. Trial design, endpoints, and potential limitations have been discussed for each trial but the authors do not think there is enough released information about each trial to include a more critical discussion.

  • Conclusion:
    - The conclusion should not only summarise the findings but also clearly state the implications for future research, clinical practice, or policy.

Response 9: A sentence about further research has been added into the conclusion, along with another sentence about potential strategies.